# Production of a rabbit monoclonal antibody for highly sensitive detection of citrus mosaic virus and related viruses

Shogo Miyoshi[1], Soh Tokunaga[1], Tatsuhiko Ozawa[2], Hiroyuki Takeda[3], Mitsuo Aono[4], Takanori Miyoshi[4], Hiroyuki Kishi[2], Atsushi Muraguchi[2], Shin-ichi Shimizu[4], Akira Nozawa[1]*, Tatsuya Sawasaki[1]*

1 Division of Cell-Free Sciences, Proteo-Science Center, Ehime University, Matsuyama, Ehime, Japan, 2 Department of Immunology, Graduate School of Medical and Pharmacological Science, University of Toyama, Toyama, Toyama, Japan, 3 Division of Proteo-Drug-Discovery Sciences, Proteo-Science Center, Ehime University, Matsuyama, Ehime, Japan, 4 Fruit Tree Research Center, Ehime Research Institute of Agriculture, Forestry and Fisheries, Matsuyama, Ehime, Japan

* nozawa.akira.my@ehime-u.ac.jp (AN); sawasaki@ehime-u.ac.jp (TS)

**Data Availability Statement:** All relevant data are within the manuscript and its Supporting Information files.

## Abstract

Citrus mosaic virus (CiMV) is one of the causal viruses of citrus mosaic disease in satsuma mandarins (*Citrus unshiu*). Prompt detection of trees infected with citrus mosaic disease is important for preventing the spread of this disease. Although rabbit monoclonal antibodies (mAbs) exhibit high specificity and affinity, their applicability is limited by technical difficulties associated with the hybridoma-based technology used for raising these mAbs. Here, we demonstrate a feasible CiMV detection system using a specific rabbit mAb against CiMV coat protein. A conserved peptide fragment of the small subunit of CiMV coat protein was designed and used to immunize rabbits. Antigen-specific antibody-producing cells were identified by the immunospot array assay on a chip method. After cloning of variable regions in heavy or light chain by RT-PCR from these cells, a gene set of 33 mAbs was constructed and these mAbs were produced using Expi293F cells. Screening with the AlphaScreen system revealed eight mAbs exhibiting strong interaction with the antigen peptide. From subsequent sequence analysis, they were grouped into three mAbs denoted as No. 4, 9, and 20. Surface plasmon resonance analysis demonstrated that the affinity of these mAbs for the antigen peptide ranged from $8.7 \times 10^{-10}$ to $5.5 \times 10^{-11}$ M. In addition to CiMV, mAb No. 9 and 20 could detect CiMV-related viruses in leaf extracts by ELISA. Further, mAb No. 20 showed a high sensitivity to CiMV and CiMV-related viruses, simply by dot blot analysis. The anti-CiMV rabbit mAbs obtained in this study are envisioned to be extremely useful for practical applications of CiMV detection, such as in a virus detection kit.

## Introduction

Satsuma dwarf virus (SDV) and SDV-related viruses cause serious diseases in satsuma mandarin (*Citrus unshiu* Marc.) [1]. The satsuma mandarin trees infected by these viruses become

**Funding:** AN, VP30418088600, Adaptable and Seamless Technology transfer Program through Target-driven R&D from Japan Science and Technology Agency, https://www.jst.go.jp/tt/EN/univ-ip/a-step.html, No TO, HT, and TS, 19am0101077, the Platform Project for Supporting Drug Discovery and Life Science Research (Basis for Supporting Innovative Drug Discovery and Life Science Research (BINDS)) from AMED, https://www.binds.jp, No TS, JP16H06579, a Grant-in-Aid for Scientific Research on Innovative Areas (JP16H06579 for T.S.) from the Ministry of Education, Culture, Sports, Science and Technology, Japan, http://www.mext.go.jp/en/, No.

**Competing interests:** The authors have declared that no competing interests exist.

dwarfed, develop boat- and/or spoon-shaped leaves, and leads to reduced sugar content in the fruits [1,2]. Infection by these viruses degrades quality of the fruits and causes serious economic burden [3]. These viruses are believed to be transmitted through grafting and soil [1,4]. At present, there are no available measures for removing these viruses from infected trees. Therefore, early detection of infected trees and their elimination are important, particularly for minimizing the economic burden associated with these viral infections.

SDV and SDV-related viruses have icosahedral virions containing a bipartite positive-sense single-stranded RNA genome (RNA1; 7.0 kb, RNA2; 5.4 kb) [1]. All virions consist of two kinds of coat proteins (large component; 42 kDa and small component; 22 kDa) encoded by RNA2 [5,6]. SDV and SDV-related viruses have been diagnosed by polymerase chain reaction (PCR) and/or enzyme linked immunosorbent assay (ELISA) [1,7]. Since PCR and ELISA need specialized techniques and expensive equipment and reagents, an immunochromatographic assay (ICA) has been developed for detection of these viruses during field samples. In particular, an ICA system developed by Kusano et al. is now commercially available and has emerged as an excellent diagnostic system due to its higher sensitivity and reliability [8]. This system can be used to diagnose not only SDV but also SDV-related viruses, such as citrus mosaic virus (CiMV) and navel orange infectious mottling virus (NIMV). However, novel CiMV isolates that are difficult to detect with this system have been recently reported [9,10]. One of these isolates, CiMV Az-1(B291), detected in Ehime prefecture, Japan has led to widespread infections and serious economic damage to satsuma mandarin cultivation [11]. A new diagnosis system for the detection of these isolates needs to be urgently developed.

To develop a diagnosis system such as ICA kit, development of highly specific and high affinity monoclonal antibody (mAb) is essential. In general, mAbs are isolated by a mouse hybridoma-based technology [12]. Rabbit mAbs are known to show high specificity and affinity [13,14]. However, hybridoma-based technology for raising rabbit mAbs is not standardized due to associated technical challenges. The antibody-secreting cell screening system using immunospot array assay on a chip (ISAAC) method has emerged as an efficient method for the rapid isolation of mAbs [15,16]. In this method, a single lymphocyte secreting the desired antibody can be isolated with ease and rapidity by detecting target cells on a microwell array chip. Since the ISAAC method does not require the generation of hybridoma cells, it is possible to obtain human and rabbit mAbs [14–16]. In this study, we isolated rabbit mAbs against CiMV Az-1(B291) by using the ISAAC method. Several rabbit mAbs that recognize CiMV were obtained by the AlphaScreen-screening system. These mAbs were further characterized with respect to their affinity and specificity for CiMV and related viruses by AlphaScreen, immunoblot, surface plasmon resonance (SPR), ELISA, and dot blot analysis.

## Materials and methods

### Plasmid construction

All primer sequences used in this study are listed in S1 Table. pEU-based expression vectors, including pEU-E01-GW, pEU-E01-bls-GW, and pEU-E01-bls-GST-GW were used for wheat germ cell-free protein synthesis. pcDNA3.4 (Thermo Fisher Scientific, San Jose, CA, USA) was used a vector for mammalian cell expression.

Antibody expression vector was constructed using inverse PCR and In-Fusion® HD Cloning Kit (Takara Bio, Otsu, Japan). Vectors for cell-free synthesis of CiMV coat protein and antigen peptide sequence-fused GST protein were made by using Gateway technology (Thermo Fisher Scientific, San Jose, CA, USA). Substitution of amino acids in the antigen peptide sequence was carried out using the PrimeSTAR® Mutagenesis Basal Kit (Takara Bio, Otsu, Japan).

## Isolation of anti-CiMV coat protein rabbit mAbs by ISAAC

Rabbit experiments were approved by the Committee on Animal Experiments at the University of Toyama (2015MED-11). We immunized 12- to 13-week-old white New Zealand rabbits (Sankyo Lab, Tokyo, Japan) subcutaneously with 0.5 mg antigen peptide conjugated with keyhole limpet hemocyanin in complete Freund's adjuvant. Two, four, six, and eight weeks after primary immunization, we boosted the rabbit subcutaneously with 1 mg antigen peptide in complete Freund's adjuvant. One week after the final boost, the rabbits were euthanized by administration of xylazine (35 mg/kg) and ketamine (5 mg/kg). Thereafter, the spleen was taken out from the rabbits. We collected cells from spleen and isolated rabbit IgG+ cells with rabbit IgG-specific antibody-conjugated microbeads (Miltenyi Biotec, Bergisch Gladbach, Germany) using an autoMACS® Pro Separator (Miltenyi Biotec, Bergisch Gladbach, Germany) according to the manufacturer's instructions. ISAAC was carried out as described in our previous report with slight modifications [17]. The surface of the ISAAC chip was treated with PBS-diluted rabbit IgG-specific antibody (MP Biomedicals, Irvine, CA, USA) for 2 h at room temperature. After blocking with 0.01% Biolipidure® (NOF Corporation, Tokyo, Japan) for 15 min at room temperature, cells were arrayed on the chip and incubated for 3 h to trap the secreted IgG. The chip was then treated with the biotinylated antigen peptide, followed by treatment with Cy3-conjugated streptavidin (Sigma-Aldrich Japan, Tokyo, Japan). After cells were stained with 1 mM Oregon Green (Molecular Probes, Eugene, OR, USA), antibody-secreting cells were collected using a micromanipulator (TransferMan NK2, Eppendorf Japan, Tokyo, Japan) under a BX51WI fluorescence microscope (Olympus, Tokyo, Japan). Isolation of mRNA and amplification of cDNA fragments of IgG were performed by reverse transcription polymerase chain reaction (RT-PCR) as described previously [15,18].

## Production of anti-CiMV coat protein antibody

cDNAs for heavy and light chains of the anti-CiMV coat protein antibody were subcloned into the pcDNA3.4 expression vector using PCR and In-Fusion Reaction. Each anti-CiMV coat protein antibody was expressed using the Expi293F Expression System (Thermo Fisher Scientific, San Jose, CA, USA) according to the manufacturer's protocol. The antibodies secreted in the culture medium were purified by Protein G Sepharose 4 Fast Flow antibody purification resin (GE Healthcare Japan, Hino, Japan), followed by buffer exchange using a PD-10 column (GE Healthcare Japan, Hino, Japan). Biotinylation of the antibody was carried out with a biotin labeling kit (Dojindo Laboratories, Masuki, Japan).

## AlphaScreen

All recombinant proteins were synthesized using a wheat germ cell-free synthesis system (Cell-free Sciences, Matsuyama, Japan). Biotinylation at the biotin ligation site (bls) was carried out enzymatically using BirA biotin ligase [19]. All AlphaScreen reactions were conducted in an Optiplate-384 titer plate (PerkinElmer Japan, Yokohama, Japan). AlphaScreen chemiluminescence signal was detected using the EnVision 2105 Multimode Plate Reader (PerkinElmer Japan, Yokohama, Japan).

Binding between the anti-CiMV coat protein antibody and antigen peptide was studied as follows. One μl of biotinylated antigen peptide was mixed with 22 μl of AlphaScreen buffer containing 100 mM Tris-HCl (pH 8.0), 0.01% Tween 20, 100 mM NaCl, 1 mg/ml BSA, 0.1 μl of streptavidin-conjugated AlphaScreen donor beads, and 0.1 μl of protein A-conjugated AlphaScreen acceptor beads. Subsequently, 2 μl of the culture supernatant was added to the mixture. After incubation at 26°C for 1 h, AlphaScreen signal was acquired.

The analysis of binding between anti-CiMV coat protein antibody and CiMV coat protein or antigen peptide sequence-fused GST protein was carried out as follows. One μl of CiMV coat protein or antigen peptide sequence-fused GST protein and 1 μl of 0.03 mg/ml anti-CiMV coat protein antibody were mixed with 10 μl of AlphaScreen buffer containing 100 mM Tris-HCl (pH 8.0), 0.01% Tween 20, 100 mM NaCl, and 1 mg/ml BSA, and incubated at 26°C for 1 h. CiMV coat protein and antigen peptide sequence-fused GST protein were biotinylated during the cell-free reaction. Subsequently, 13 μl of detection mixture containing 0.1 μl of streptavidin-conjugated AlphaScreen donor beads and 0.1 μl of protein A-conjugated AlphaScreen acceptor beads in AlphaScreen buffer was added to the mixture. After incubation at 26°C for 1 h, AlphaScreen chemiluminescence signal was measured.

## BIAcore assay

SPR experiments were conducted on a BIAcore X100 system (GE Healthcare Japan, Hino, Japan). HBS-EP+ (10 mM Hepes-NaOH (pH 7.4), 150 mM NaCl, 0.05% Tween 20, and 3 mM EDTA) was used as a running buffer. The temperature of the flow cells was kept at 25°C during the assay. To determine affinity between the antigen peptide and mAbs, we conducted a single-cycle kinetics assay. Concentration of Protein G-purified antibodies was determined with NanoDrop spectrophotometer (Thermo Fisher Scientific, San Jose, CA, USA). The biotinylated antigen peptide was immobilized on Sensor Chip SA (GE Healthcare Japan, Hino, Japan) at less than 3 RU. Then the antibody was injected at a range of concentrations. Flow rate was 30 μl/min, contact time was 120 sec, and dissociation time was 180 sec. The affinity parameter was calculated by using BiaEvaluation software.

## Enzyme-linked immunosorbent assay

Virus samples (CiMV Az-1(B291), SDV S-58, and NIMV NI-1) were prepared from leaves of cape lemon trees that virus infection is maintained in Fruit tree research center, Ehime research institute of agriculture, forestry, and fisheries. The existence of viruses in samples was confirmed by RT-PCR analysis according to previously described method [7]. As a negative control, we also prepared a virus-free sample from non-infected leaves. Nunc-Immuno™ MicroWell™ 96 well solid plates (Sigma-Aldrich Japan) were used. Virus samples (100 μl) were injected to wells of the micro plates and incubated for 16 h at 4°C. After blocking with 50 μl of skim milk in PBS with 0.2% Tween 20, 50 μl of antibody (0.01mg/ml) was added and incubated for 16 h at 4°C. Antibodies bound to the plate was detected by HRP-conjugated anti-rabbit IgG antibody.

## Dot blot analysis

One μl of leaf extracts was spotted onto polyvinylidene difluoride membranes and dried for 5 min. As a negative control, virus-free sample was also spotted. After blocking with 5% skim milk solubilized in a TBST buffer for 1 h, the membrane was treated for 1 h by biotinylated anti-CiMV coat protein mAb (No. 20, 0.001 μg/ml). After three times washing with TBST buffer, the membrane was treated with HRP-conjugated anti-biotin antibody. After the membrane was washed with TBST buffer three times, the antibody was detected with ImmunoStar LD using an ImageQuant LAS 4000 imager.

## Statistical analyses

We used Student's t-*t*est and Tukey test for a two-group comparison and a multiple group comparison, respectively, using a Microsoft Excel (Microsoft, Redmond, WA). Statistical significance was accepted at $P < 0.05$.

## Results

### Isolation of rabbit anti-CiMV mAbs by ISAAC

To obtain anti-CiMV mAbs, we immunized rabbits with a peptide that corresponds to a residue (11 aa) from 475 to 485 in the CiMV coat protein [20]. This region is conserved among the related viruses and predicted to form an outer region of the coat protein. In this study, we selected rabbits for mAb production because rabbit antibodies have high affinity and can be a powerful tool for diagnosis [14]. After sacrifice, anti-CiMV peptide-specific antibody-secreting cells were isolated using immunospot array assay on a chip (ISAAC). By single-cell RT-PCR, 33-pair fragments of the variable region sequences, heavy chain and light chain, were amplified. These fragments were introduced into pcDNA3.4 expression vector, and then corresponding antibodies were produced using Expi293F cultured cells. To evaluate specificity of the mAbs with CiMV peptide, we used AlphaScreen system. In this system, AlphaScreen beads generate chemiluminescence signal when a mAb binds to the CiMV peptide (Fig 1B). Among 33 antibodies, eight mAbs interacted with the CiMV peptide, which was confirmed by AlphaScreen (Fig 1C). These eight mAbs were selected for analysis of the binding ability with the CiMV coat protein synthesized using the cell-free system by AlphaScreen. As shown in Fig 1D, interaction between all eight mAbs and CiMV coat protein was observed. Thus, we analyzed sequences of these mAbs. The results showed that the eight mAbs are classified into three groups. Nucleotide sequences encoding variable region of mAbs No. 3, 4, 10, and 23 and mAbs No. 9, 22, and 33 were identical, respectively, and that of mAb No. 20 was unique. Hence, we selected mAb No. 4, 9, and 20 as representative of each group for further analysis. The interaction of the three mAbs with CiMV coat protein was further tested by immunoblot analysis, but only weak signals were detected (S1 Fig).

### Kinetic assay of anti-CiMV mAbs

To investigate the affinity of the three mAbs obtained in this study toward the CiMV coat protein, we determined their affinity toward the CiMV peptide using surface plasmon resonance (SPR). When 3 RU of biotinylated CiMV peptide was captured on a sensor chip, kinetic analysis from 3 RU yielded $K_D$ values (Fig 2). $K_D$ value of mAb No. 4, 9, and 20 was $8.7 \times 10^{-10}$ M ($K_a = 5.5 \times 10^5$ 1/Ms, $K_d = 4.8 \times 10^{-4}$ 1/s), $8.2 \times 10^{-11}$ M ($K_a = 5.0 \times 10^6$ 1/Ms, $K_d = 4.1 \times 10^{-4}$ 1/s), and $5.5 \times 10^{-11}$ M ($K_a = 4.7 \times 10^6$ 1/Ms, $K_d = 2.6 \times 10^{-4}$ 1/s), respectively. Comparing among these mAbs, although there were small differences in the dissociation rate constant ($K_d$), the association rate constant ($K_a$) of mAb No. 9 and 20 was one order higher than that of mAb No. 4. Hence, $K_D$ value of mAb No. 9 and 20 was one order lower than that of mAb No. 4. However, even a $K_D$ value of mAb No. 4 was sufficiently low. These results indicate that all three mAbs obtained in this study have high affinity toward the CiMV peptide.

Kinetics assay was conducted by single cycle analytics. The biotinylated antigen peptide was immobilized on Sensor Chip SA at less than 3 RU. Then, the purified mAb was injected at a series of concentrations. The blue lines represent the real time binding data generated from the injected mAbs while the black lines indicate the global fit of a 1:1 interaction model to each kinetic data set.

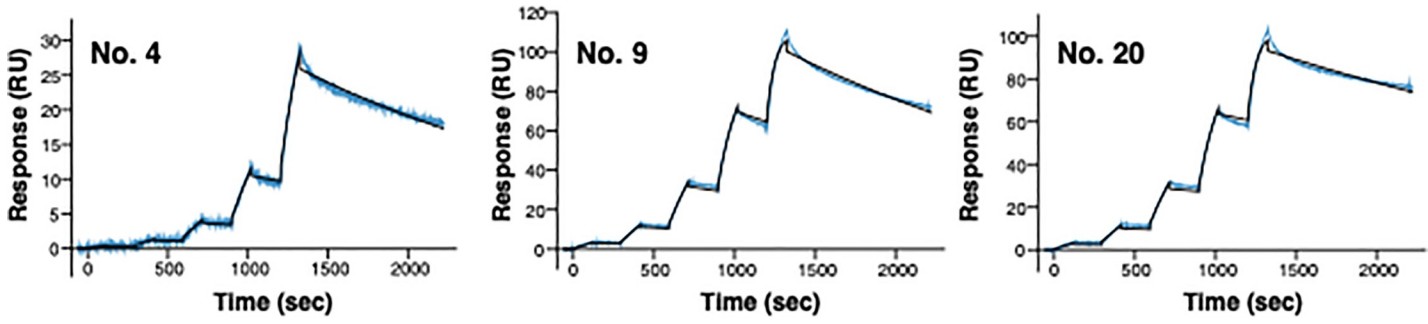

**Fig 1. Isolation of rabbit mAbs against conserved peptide in CiMV coat protein.** (A) Flow chart of experimental design. (B) Schematic diagram of antibody screening by AlphaScreen. When antibody binds to the peptide as illustrated, AlphaScreen beads trigger chemiluminescence signals. (C) Screening of mAbs against CiMV coat protein by AlphaScreen. Interaction between mAb and biotinylated peptide fragment of CiMV coat protein small subunit was analyzed by AlphaScreen. Biotinylated DHFR was used as negative control. The data are shown as means from two independent experiments. (D) Interaction analysis of mAb with CiMV coat protein by AlphaScreen. Interaction between eight mAbs and biotinylated full-length CiMV coat protein small subunit synthesized by cell-free system was analyzed by AlphaScreen. Biotinylated DHFR was used as negative control. The data are shown as means ± standard deviations (indicated with error bars) from three independent experiments. Asterisk indicates significant differences from negative control (** $P < 0.01$; *** $P < 0.001$, Student's $t$-test).

**Fig 2. Kinetic assay of anti-CiMV mAbs by SPR.**

## Specificity of anti-CiMV mAbs

CiMV belongs to a group of SDV-like viruses [21]. The group consists of Satsuma dwarf virus (SDV), CiMV, Navel orange infectious virus (NIMV), and Hyuganatsu virus (HV) [7]. To investigate the specificity of anti-CiMV mAbs isolated in this study, we analyzed cross reactivity of these mAbs with SDV-like viruses. For this purpose, interaction of these mAbs with SDV-like virus peptide-fused GST proteins was analyzed by AlphaScreen and immunoblot analysis. The CiMV sequence used as an antigenic peptide or the corresponding region of each SDV-like virus was fused to the C-terminus of GST and the SDV-like virus peptide-fused GST proteins were synthesized by the cell-free system. GST-fused sequences of each SDV-like virus are shown in Fig 3A. As shown in Fig 3B, although mAb No. 9 and 20 interacted with all four peptide sequences, mAb No. 4 weakly interacted with only HV peptide. Further, mAb No. 4 and 9 recognized all four peptide sequences in immunoblot analysis, but mAb No. 20 detected only CiMV and SDV peptide sequences (S2 Fig).

## Detection of CiMV and SDV-like viruses in infected leaves by anti-CiMV mAbs

To confirm whether mAbs obtained in this study detect CiMV and SDV-like viruses in leaves, the mAbs were evaluated by ELISA and dot blot analysis. For this analysis, leaves of virus-infected cape lemon (*Citrus jambhiri*) were used, because these viruses can easily amplify in the leaves without growth inhibition. Fig 4A shows photographs of the leaves; the trees were infected with viruses, but symptoms of infection were hardly seen in the leaves. Half of the leaves were used for RT-PCR and extracts from the remaining half were used for ELISA and dot blot analysis. Prior to the experiments, the presence of viruses in these leaves was confirmed by RT-PCR analysis. PCR bands from viruses were detected after 50 cycles in samples from virus-infected leaves, while no band was seen in samples from non-infected leaves (Fig 4B). The amplified bands were sequenced, and the sequence of each sample was confirmed to correspond to that of the expected virus. Using the remaining half of the leaf samples, we detected viruses by ELISA. The results showed that signals were significantly higher in the CiMV and NIMV sections of mAb No. 9 and 20 whereas equivalent or weaker signals were observed in the SDV section of mAb No. 9 and 20 (Fig 4C). In all sections of mAb No. 4, equivalent or weaker signals comparing with virus -free sample were observed (Fig 4C). These results demonstrate that mAb No. 9 and 20 are applicable for CiMV and NIMV detection by ELISA.

For SDV-related viruses, a highly sensitive diagnosis kit has been developed [8]. By using this kit, we tried to detect SDV, NIMV and the CiMV Az-1(B291) isolate. In fact, this kit detected SDV (S3 Fig). However, NIMV and the CiMV Az-1(B291) isolate could not be recognized, indicating the importance of a novel mAb to detect the infection of NIMV and the CiMV Az-1(B291) isolate.

To develop a simplified detection method using dot blot analysis, mAb No. 20 was selected and biotinylated. Diluted samples from leaf extracts were spotted on to the polyvinylidene difluoride membrane. Viruses on these membranes were detected using biotinylated mAb No. 20 and HRP-conjugated anti-biotin antibody as primary and secondary antibodies, respectively. As shown in Fig 4D, signals were detected in all three virus samples even in 100-fold diluted spots. This result indicates that mAb No. 20 has sufficient sensitivity for the simple detection of CiMV, NIMV, and SDV in infected citrus leaves by dot blot analysis.

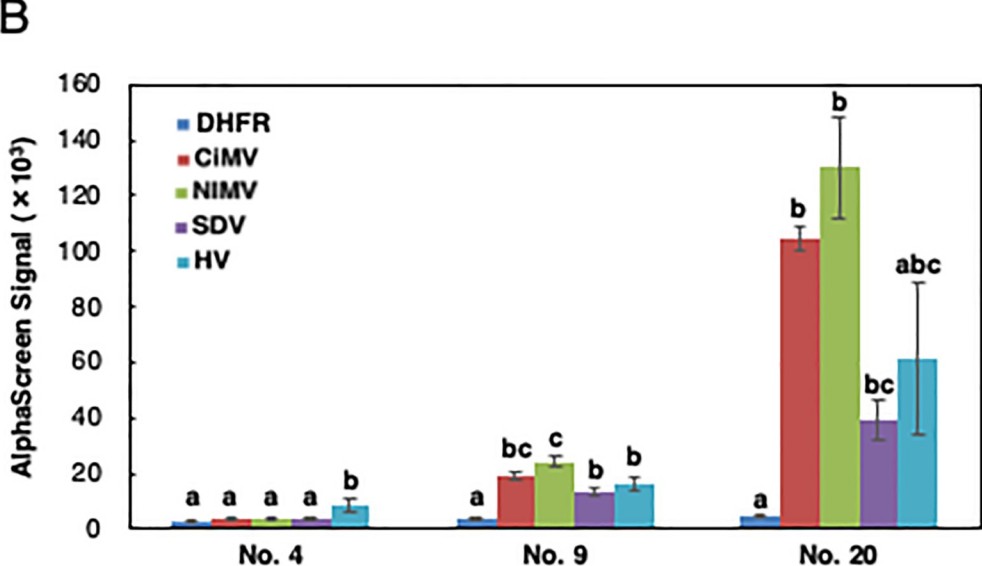

**Fig 3. Specificity of mAbs against CiMV and related viruses.** (A) Peptide sequences of CiMV and related viruses. Sequence of CiMV represents a part of the coat protein small subunit used as peptide antigen for antibody production. Regions corresponding to CiMV coat protein sequence in other viruses are also shown. Red letters indicate amino acid residues dissimilar to CiMV sequence. Corresponding amino-acid residue number of the first H of each peptide in each coat protein is shown. (B) Interaction analysis of mAbs with SDV-like virus peptide-fused GST proteins by AlphaScreen. Three mAbs and four biotinylated SDV-like virus peptide-fused GST proteins synthesized by the cell-free system were used. Biotinylated DHFR was used as negative control. The data are shown as means ± standard deviations (indicated with error bars) from three independent experiments. Different letters indicate significant differences ($P < 0.05$, Tukey test).

## Discussion

In this study, we obtained rabbit mAbs suitable for virus detection, using a peptide as an antigen by the ISAAC method. Although it is generally difficult to obtain high affinity antibodies using peptides as antigens, our results showed that it is possible to obtain mAbs for practical use using a peptide consisting of as few as 11 amino acid residues by the ISAAC method. Combinatorial utilization of a peptide and the ISAAC method allows obtaining mAbs against proteins that are hard to obtain in sufficient amounts for immunization.

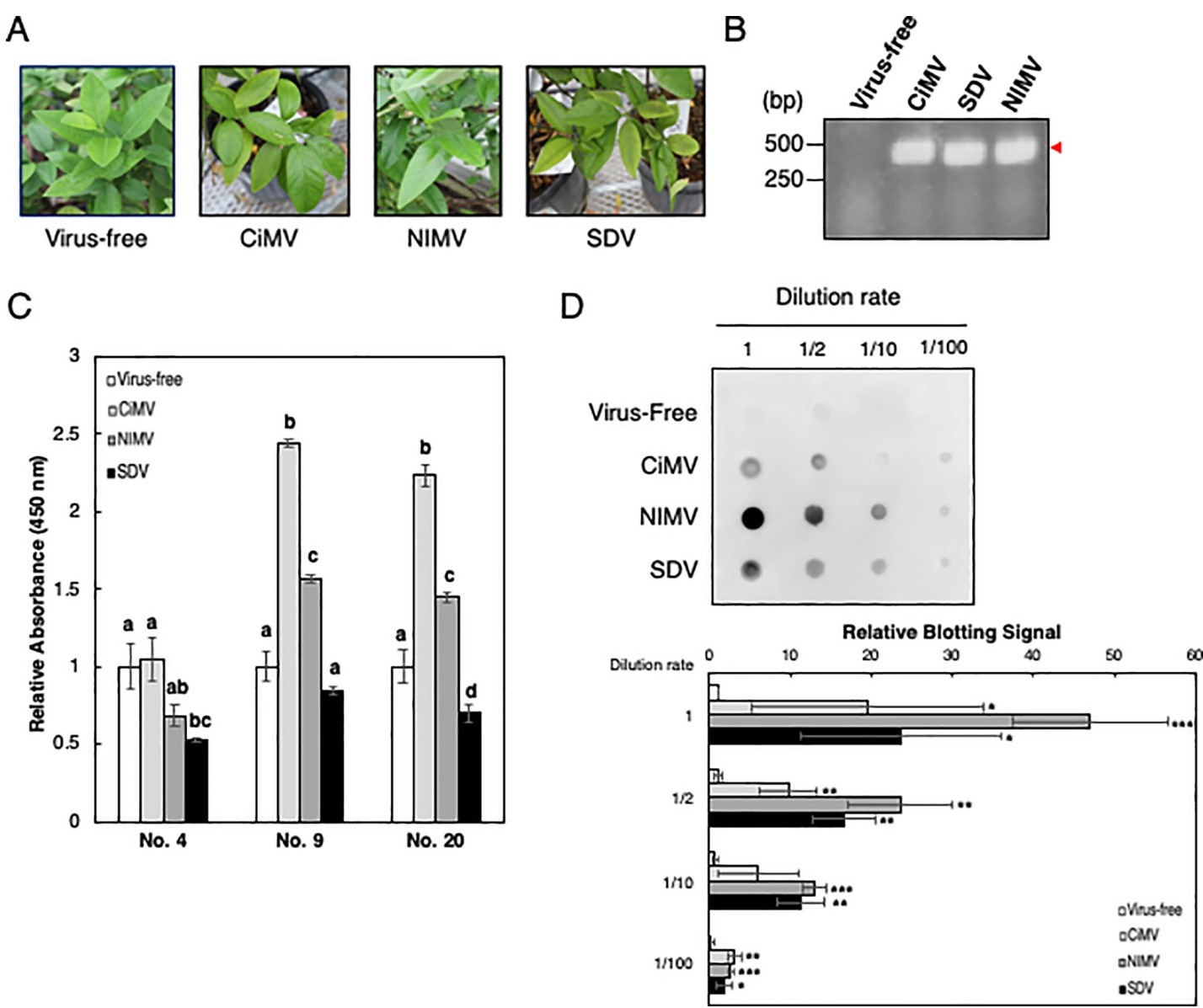

**Fig 4. Detection of viruses in plant leaves with mAbs.** (A) Rough lemon infected with virus. Leaves from these plants were used in following assays. (B) Detection of viruses by RT-PCR. RNA was extracted from leaves of virus-infected rough lemon and subjected to RT-PCR. Arrowhead indicates obtained amplicons. (C) Detection of viruses by ELISA. Proteins were extracted from leaves of virus-infected rough lemon and were coated onto 96-well plates. The three mAbs (0.01 mg/ml) and HRP-linked anti-rabbit IgG antibody (0.02 µg/ml) were used as primary and secondary antibodies, respectively. The data are shown as means ± standard deviations (indicated with error bars) from three independent experiments. Different letters indicate significant differences ($P < 0.05$, Tukey test). (D) Detection of viruses in extracts of leaves by dot blot. Proteins were extracted from leaves of virus-infected rough lemon and were spotted onto polyvinylidene difluoride membranes. Biotinylated mAb no. 20 (0.001 mg/ml) was used as primary antibody and HRP-linked anti-biotin antibody (0.1 µg/ml) was used as secondary antibody. The representative result is shown. Spot signals from three independent experiments were quantitated using Image J and the obtained signal values were converted to relative values against the signal value of non-diluted virus-free samples. These normalized data were shown as means ± standard deviations (indicated with error bars). Asterisk indicates significant differences from virus-free controls (* $P < 0.05$; ** $P < 0.01$; *** $P < 0.001$, Student's $t$-test).

Three kinds of anti-CiMV mAbs with different variable regions were obtained in this study. These mAbs showed higher interaction with antigenic peptides and proteins in AlphaScreen, SPR analysis, and ELISA (Figs 1C, 2, 3B and 4C), but only weak signals were detected in immunoblots (S1 Fig and S2 Fig). These results suggested that the mAbs are conformation-sensitive. This type of antibodies is reported to be hard to recognize unfolded proteins [22].

Among the three mAbs obtained in this study, mAb No. 4 showed low interaction with GST-fused CiMV coat protein fragment (Fig 3B) and native CiMV virus (Fig 4C). These results indicate that mAb No. 4 might recognize the amino terminus-containing region of the CiMV antigenic peptide. On the other hand, the remaining two mAbs, mAb No. 9 and 20, showed high interaction activity with GST-fused CiMV coat protein fragment (Fig 3B) and native CiMV virus (Fig 4C and 4D). Furthermore, the two mAbs recognized not only CiMV, but also SDV-like viruses (Figs 3B, 4C and 4D). These results indicate that mAb No. 9 and 20 can potentially be used to create a virus detection kit based on ICA. The detection kit would contribute to the development of a diagnosis system against CiMV and its related viruses.

In ELISA and dot blot analysis, we used the CiMV Az-1(B291) isolate. This isolate recently caused serious damage to citrus farms in the Ehime prefecture, Japan. Although amino-acid sequence of coat protein of SDV (S-58) is 20% different with that of CiMV (ND-1), the CiMV Az-1(B291) isolate has further 3.3% different amino-acid sequence of coat protein with CiMV (ND-1). Although a highly sensitive diagnosis kit for SDV-related viruses has been developed [8], it could not detect NIMV and the CiMV Az-1(B291) isolate (S3 Fig). On the other hand, the results of ELISA and dot blot analysis (Fig 4C and 4D) clearly show that the mAbs obtained in this study recognize the CiMV Az-1(B291) isolate. By utilizing these mAbs, we expect that a sensitive and simple diagnosis kit against the CiMV Az-1(B291) isolate could be developed.

## Supporting information

**S1 Fig. Detection of CiMV coat protein by immunoblotting.** Three mAbs were used for detection of biotinylated GST-fused CiMV coat protein fragment by immunoblotting. Arrowheads indicate GST-fused CiMV coat protein fragment.
(TIFF)

**S2 Fig. Detection of different types of viruses by immunoblotting.** Three mAbs were used for detection of four different biotinylated SDV-like virus peptide-fused GST proteins synthesized using the cell-free system. Arrowheads indicate SDV-like virus peptide-fused GST proteins.
(TIFF)

**S3 Fig. Detection of different types of viruses by a virus detection kit.** Leaves of virus-infected rough lemon were applied to detection assay. Detection of viruses using an SDV detection kit (SDV Chromato, Mizuho Medy Co., Ltd., Tosu, Japan) was performed according to the manufacturer's instruction.
(TIFF)

**S1 Table. Primer sequences.**
(XLSX)

**S1 File. Method of immunoblot analysis.**
(DOCX)

**S1 Raw image.**
(PDF)

## Acknowledgments

We thank all staffs of Fruit Tree Research Center and The Orange Research Center (Ehime Research Institute of Agriculture, Forestry and Fisheries) for plant managements, C. Takahashi

and C. Furukawa for technical assistance, and the Applied Protein Research Laboratory of Ehime University for protein analysis.

## Author Contributions

**Data curation:** Akira Nozawa.

**Funding acquisition:** Tatsuhiko Ozawa, Hiroyuki Takeda, Akira Nozawa, Tatsuya Sawasaki.

**Investigation:** Shogo Miyoshi, Soh Tokunaga, Tatsuhiko Ozawa, Hiroyuki Takeda, Mitsuo Aono, Takanori Miyoshi, Hiroyuki Kishi, Atsushi Muraguchi, Shin-ichi Shimizu, Akira Nozawa.

**Project administration:** Tatsuya Sawasaki.

**Supervision:** Shin-ichi Shimizu, Akira Nozawa, Tatsuya Sawasaki.

**Writing – original draft:** Shogo Miyoshi, Akira Nozawa, Tatsuya Sawasaki.

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
