## [Decision Letter · Decision Letter 0]

26 Sep 2019

PONE-D-19-23810

Production of a rabbit monoclonal antibody for highly sensitive detection of citrus mosaic virus and related viruses

PLOS ONE

Dear Akira Nozawa,

Thank you for submitting your manuscript to PLOS ONE. After careful consideration, we feel that it has merit but does not fully meet PLOS ONE’s publication criteria as it currently stands. Therefore, we invite you to submit a revised version of the manuscript that addresses the points raised during the review process.

We would appreciate receiving your revised manuscript by Nov 10 2019 11:59PM. To enhance the reproducibility of your results, we recommend that if applicable you deposit your laboratory protocols in protocols.io, where a protocol can be assigned its own identifier (DOI) such that it can be cited independently in the future. For instructions see: http://journals.plos.org/plosone/s/submission-guidelines#loc-laboratory-protocols

We look forward to receiving your revised manuscript.

Kind regards,

Gourapura J Renukaradhya

Academic Editor

PLOS ONE

Journal Requirements:

2. To comply with PLOS ONE submission requirements regarding animal research https://journals.plos.org/plosone/s/submission-guidelines#loc-animal-research, in your Methods section, please provide additional information regarding the experiments involving rabbits and clarify whether rabbits are scarified in your laboratory and if so, ensure you have included details on (1) methods of sacrifice, (2) methods of anesthesia and/or analgesia, and (3) efforts to alleviate suffering.

Reviewers' comments:

Reviewer's Responses to Questions

**Comments to the Author**

1. Is the manuscript technically sound, and do the data support the conclusions?

Reviewer #1: Partly

Reviewer #2: Partly

2. Has the statistical analysis been performed appropriately and rigorously? 

Reviewer #1: No

Reviewer #2: No

3. Have the authors made all data underlying the findings in their manuscript fully available?

Reviewer #1: Yes

Reviewer #2: Yes

4. Is the manuscript presented in an intelligible fashion and written in standard English?

Reviewer #1: Yes

Reviewer #2: Yes

5. Review Comments to the Author

Reviewer #1: 1. The manuscript is technically sound and the data support conclusions partially. However there are sections where it needs improvement and they are listed below.

a. In the Results section, in Isolation of rabbit anti-CiMV mAbs by ISAAC, in the last paragraph, you mention "Thus, we analyzed sequences of these mAbs. The results showed that the eight mAbs are classified into three groups." The rationale for this classification is not clear.

b.Kinetic assay of anti-CiMV mAbs is not clearly explained. There are differences in the affinities of anti-CiMV mAbs, but they all exhibit 1:1 interaction model. What could be the reason behind this observation?

c.The quality of immunoblots should improve.

d. In Detection of CiMV and SDV-like viruses in infected leaves by anti-CiMV mAbs section, in the line 309, you state " In the SDV section of mAb No.9 and 20, early equivalent signals were observed". This statement is wrong. Here, the true signal is zero as it falls in the background noise.

e. In the Discussion section, in the line 358, you mention "Although a highly sensitive diagnostic kit for SDV-related viruses has been developed, the CiMV Az-1(B291) isolate has still been hard to detect". This kit should be used as a positive control in your experiments and you need to demonstrate how your novel rabbit anti-CiMV mAbs are superior to this kit.

2. The statistical analyses have not been performed appropriately and rigorously. All the experiments should be performed multiple number of times and expressed as mean plus or minus standard error of the mean. The sample size should be adequate and this should be mentioned clearly in the figure legends. The immunoblots should be quantified and depicted. Appropriate negative and positive controls should be used.

3. All the data is fully available.

4.Overall, the manuscript is presented in an intelligible fashion and written in standard English.

Reviewer #2: Production of a rabbit monoclonal antibody for highly sensitive detection of citrus mosaic virus and related viruses by Miyoshi et al (Manuscript ID #PONE-D-19-23810; PLOS ONE).

The authors have reported characterization of rabbit monoclonal antibody for sensitive detection of citrus mosaic virus (CiMV) and related viruses. The manuscript is well written with proper introduction and need for this study. However, there are several shortcomings as listed below.

1. Have you compared the sensitivity of commercially available kit (by Kusano et al) against novel CiMV isolates in your studies? Data has to be included to compare sensitivity of the mAbs identified in this paper.

2. How similar are these viruses with respect to viral genome (CiMV, novel CiMV isolates and SDV-related viruses), discuss in the paper.

3. What is the difference between Fig 1C and 1D? Did you include negative control (DHFR) in Fig 1C? Fig 1 legend is not clear on CiMV coat protein/ peptide, needs to be modified.

4. Fig 1E, although there is corresponding staining as indicated by arrowheads, there are additional bands – is it expected?

5. AlphaScreen data shows differential sensitivity for the various peptides from SDV-like viruses, Fig 3B. Is there a statistical difference between test groups vs. negative control (DHFR)? Also, what does the error bars means – include details in the figure legend.

6. Include corresponding amino acid residue numbers in Fig 3A.

7. It is not clear about the specificity mAbs in Fig 3C, except for mAb No. 9 (Lines 277 to 279).

8. Provide reference for statements in the Introduction section (lines 67 to 70).

9. Perform statistical analysis for the ELISA data in Fig 4C, provide details on number of experiments including replicates involved.

10. How many experiments involved in Fig 4D? Provide quantitated dot-blot analysis based on staining density – this analysis help better understand sensitivity of mAb No. 20.

6. PLOS authors have the option to publish the peer review history of their article (what does this mean?). If published, this will include your full peer review and any attached files.

Reviewer #1: No

Reviewer #2: No

---

## [Author Response · Author response to Decision Letter 0]

30 Oct 2019

Point-by-Point Responses to the Reviewers’ Critiques (PONE-D-19-23810)

We deeply appreciate the thorough analysis and constructive suggestions provided by the two reviewers to further improve our manuscript. As described in more detail below, we have experimentally addressed all the reviewers’ concerns. With this extensive revision, we hope that the reviewers will concur with us that we have addressed all of the raised concerns in a satisfactory manner and, consequently, substantially strengthened our paper.

Reviewer comments: 

Reviewer #1 (Remarks to the Author): 

The manuscript is technically sound and the data support conclusions partially. However there are sections where it needs improvement and they are listed below.

We wish to express our appreciation to the reviewer for his/her insightful comments, which have helped us significantly improve the paper.

Comment 1.

In the Results section, in Isolation of rabbit anti-CiMV mAbs by ISAAC, in the last paragraph, you mention "Thus, we analyzed sequences of these mAbs. The results showed that the eight mAbs are classified into three groups." The rationale for this classification is not clear.

Answer 1.

We thank the reviewer for bringing up these constructive comments. We added the following sentences in the results section (Line 224-227).

“Nucleotide sequences encoding variable region of mAbs No. 3, 4, 10, and 23 and mAbs No. 9, 22, and 33 were identical, respectively, and that of mAb No. 20 was unique. Hence, we selected mAbs No. 4, 9, and 20 as representative of each group for further analysis.”

Comment 2.

Kinetic assay of anti-CiMV mAbs is not clearly explained. There are differences in the affinities of anti-CiMV mAbs, but they all exhibit 1:1 interaction model. What could be the reason behind this observation?

Answer 2.

We thank you for your thoughtful comments for the improvement of our manuscript. In our experimental condition, biotinylated antigen peptides were immobilized on sensor chip at very low concentration (less than 3 RU). As interaction mode between antigen peptides and mAbs was expected to become 1:1 binding, we evaluated the results with 1:1 interaction model. 

In response to the reviewer’s comment, we added following sentence as an explanation of affinity among mAbs in results section (Line 251-255).

“Comparing among these mAbs, although there were small difference in the dissociation rate constant (Kd), the association rate constant (Ka) of mAb No. 9 and 20 was one order higher than that of mAb No. 4. Hence, KD value of mAb No. 9 and 20 was one order lower than that of mAb No. 4. However, KD value of mAb No. 4 was sufficiently low.”

In addition, we rewrote the explanation of experimental condition in legend as followings (Line 260-263).

“Then, the purified mAb was injected at a series of concentrations. The blue lines represent the real time binding data generated from the injected mAbs while black lines indicate the global fit of a 1:1 interaction model to each kinetic data set.”

Comment 3.

The quality of immunoblots should improve.

Answer 3.

We sincerely thank the reviewer for raising this concern. As commented in Line 355-358 in discussion session, the three mAbs obtained in this study look to be structure-recognition antibody. This type of antibodies can not recognize unfolded proteins (Matsuoka et al., 2010). So, we think only weak signals were detected by immunoblot analyses. We moved the immunoblot data (Fig 1E and 3C) to supporting information. In addition, we added following sentence in discussion section (Line 358-359) for explanation of structure-recognition antibody.

“This type of antibodies is reported to be hard to recognize unfolded proteins [22].”

22. Matsuoka K, Komori H, Nose M, Endo Y, Sawasaki T. Screening method for autoantigen proteins using the biotinylated protein library produced by wheat cell-free synthesis. J Proteome Res 2010; 9: 4264-4273.

Comment 4.

In Detection of CiMV and SDV-like viruses in infected leaves by anti-CiMV mAbs section, in the line 309, you state " In the SDV section of mAb No.9 and 20, early equivalent signals were observed". This statement is wrong. Here, the true signal is zero as it falls in the background noise.

Answer 4. 

We thank the reviewer for pointing out this critical issue and would like to apologize for our wrong explanation. In revised manuscript, we replaced the sentences “The result showed that signals were 1.5 to 2.5 times higher in the CiMV and NIMV sections of mAbs No. 9 and 20 (Fig 4C) whereas equivalent or weaker signals were observed in all sections of the No. 4 mAb. In the SDV section of mAb No. 9 and 20, early equivalent signals were observed.” with “The result showed that signals were significantly higher in the CiMV and NIMV sections of mAbs No. 9 and 20 whereas equivalent or weaker signals were observed in the SDV section of mAb No. 9 and 20 (Fig 4C). In all sections of mAb No. 4, equivalent or weaker signals comparing with virus-free sample were observed (Fig 4C).” in the results section (Line 306-310).

Comment 5.

In the Discussion section, in the line 358, you mention "Although a highly sensitive diagnostic kit for SDV-related viruses has been developed, the CiMV Az-1(B291) isolate has still been hard to detect". This kit should be used as a positive control in your experiments and you need to demonstrate how your novel rabbit anti-CiMV mAbs are superior to this kit.

Answer 5.

We thank you for your thoughtful comments for the improvement of our manuscript. In response to the reviewer’s comment, we performed detection of SDV and CiMV by using the diagnosis kit. The results are shown in S3 Fig. In addition, we added following sentences for explanation of the result of the experiment in results section (Line 312-316) and discussion section (Line 373-377). 

“For SDV-related viruses, a highly sensitive diagnosis kit for SDV-related viruses has been developed [8]. By using the kit, as tried to detect SDV, NIMV and the CiMV Az-1(B291) isolate. In fact, this kit detected SDV (S3 Fig). However, NIMV and the CiMV Az-1(B291) isolate could not be recognized, indicating the importance of a novel mAb to detect the infection of NIMV and the CiMV Az-1(B291) isolate.”

“Although a highly sensitive diagnosis kit for SDV-related viruses has been developed [8], it could not detect NIMV and the CiMV Az-1(B291) isolate (S3 Fig). On the other hand, the results of ELISA and dot blot analysis (Fig 4C and D) clearly show that the mAbs obtained in this study recognize the CiMV Az-1(B291) isolate.”

Comment 6.

The statistical analyses have not been performed appropriately and rigorously. All the experiments should be performed multiple number of times and expressed as mean plus or minus standard error of the mean. The sample size should be adequate and this should be mentioned clearly in the figure legends. The immunoblots should be quantified and depicted. Appropriate negative and positive controls should be used.

Answer 6.

We thank the reviewer for bringing up these constructive comments. In response to the reviewer’s comment, we performed multiple dot-blot analysis and the results were quantitated (Fig 4D). In addition, we added data of negative control in Fig 1C and performed statistical analysis in Fig 1D, 3B, 4C, and 4D.

 

Reviewer #2 (Remarks to the Author): 

The authors have reported characterization of rabbit monoclonal antibody for sensitive detection of citrus mosaic virus (CiMV) and related viruses. The manuscript is well written with proper introduction and need for this study. However, there are several shortcomings as listed below.

We wish to express our appreciation to the reviewer for his/her kind comments, which have helped us significantly improve the paper. 

Comment 1.

Have you compared the sensitivity of commercially available kit (by Kusano et al) against novel CiMV isolates in your studies? Data has to be included to compare sensitivity of the mAbs identified in this paper.

Answer 1.

We thank you for your thoughtful comments for the improvement of our manuscript. In response to the reviewer’s comment, we performed detection of SDV and CiMV by using the diagnosis kit. The results are shown in S3 Fig. In addition, we added following sentences for explanation of the result of the experiment in results section (Line 312-316) and discussion section (Line 373-377). 

“For SDV-related viruses, a highly sensitive diagnosis kit for SDV-related viruses has been developed [8]. By using the kit, as tried to detect SDV, NIMV and the CiMV Az-1(B291) isolate. In fact, this kit detected SDV (S3 Fig). However, NIMV and the CiMV Az-1(B291) isolate could not be recognized, indicating the importance of a novel mAb to detect the infection of NIMV and the CiMV Az-1(B291) isolate.”

“Although a highly sensitive diagnosis kit for SDV-related viruses has been developed [8], it could not detect NIMV and the CiMV Az-1(B291) isolate (S3 Fig). On the other hand, the results of ELISA and dot blot analysis (Fig 4C and D) clearly show that the mAbs obtained in this study recognize the CiMV Az-1(B291) isolate.”

Comment 2.

How similar are these viruses with respect to viral genome (CiMV, novel CiMV isolates and SDV-related viruses), discuss in the paper.

Answer 2. 

We thank you for your thoughtful comments for the improvement of our manuscript. In response to the reviewer’s comment, we added following sentences in discussion section (Line 370-373).

“Although amino acid sequence of coat protein of SDV (S-58) is 20% different with that of CiMV (ND-1), the CiMV Az-1(B291) isolate has further 3.3% different amino acid sequence of coat protein with CiMV (ND-1).”

Comment 3.

What is the difference between Fig 1C and 1D? Did you include negative control (DHFR) in Fig 1C? Fig 1 legend is not clear on CiMV coat protein/ peptide, needs to be modified.

Answer 3. 

We thank you for your thoughtful comments for the improvement of our manuscript. Fig 1C and 1D showed results of interaction assay of mAbs with peptide and full-length protein, respectively. We added data of negative control using DHFR to Fig 1C. In addition, we performed statistical analysis in Fig 1D. For better understanding, we replaced legend of Fig 1C and 1D as described below (Line 233-242).

“(C) Screening of mAbs against CiMV coat protein by AlphaScreen. Interaction between mAb and biotinylated peptide fragment of CiMV coat protein small subunit was analysed by AlphaScreen. Biotinylated DHFR was used as negative control. The data are shown as means from two independent experiments. (D) Interaction analysis of mAb with CiMV coat protein by AlphaScreen. Interaction between eight mAbs and biotinylated full-length CiMV coat protein small subunit synthesized by cell-free system was analysed by AlphaScreen. Biotinylated DHFR was used as negative control. The data are shown as means ± standard deviations (indicated with error bars) from three independent experiments. Asterisk indicates significant differences from negative control (**P < 0.01; ***P < 0.001, Student’s t-test).”

Comment 4.

Fig 1E, although there is corresponding staining as indicated by arrowheads, there are additional bands – is it expected?

Answer 4.

We sincerely thank the reviewer for raising this concern. As we explained in manuscript (Lines 355-358), all mAbs we obtained in this study are looked like structure-recognition antibody. So, it is hard to detect denatured proteins by these mAbs. We think additional bands are non-specific bands derived from wheat germ extract. We removed following sentences and Fig 1E was moved to supporting information. With this decision, procedure of immunoblot analysis was also moved to supporting information. 

“The interaction of the three mAbs with CiMV coat protein was also confirmed by immunoblot analysis (Fig 1E).”

Comment 5. 

AlphaScreen data shows differential sensitivity for the various peptides from SDV-like viruses, Fig 3B. Is there a statistical difference between test groups vs. negative control (DHFR)? Also, what does the error bars means – include details in the figure legend. 

Answer 5.

We thank you for your thoughtful comments for the improvement of our manuscript. According to this suggestion, we performed statistical analysis in Fig 3B. For better understanding, we inserted following sentences in legend of Fig 3B (Line 288-291).

“Biotinylated DHFR was used as negative control. The data are shown as means ± standard deviations (indicated with error bars) from three independent experiments. Different letters indicate significant differences (P < 0.05, Tukey test).”

Comment 6.

Include corresponding amino acid residue numbers in Fig 3A.

Answer 6.

We thank you for your thoughtful comments for the improvement of our manuscript. According to this suggestion, we described number of amino acid residues in Fig 3A and added following sentence about amino acid residue numbers in legend of Fig 3A (Line 284-285). 

“Corresponding amino acid residue number of first H of each peptide in each coat protein is shown.”

Comment 7.

It is not clear about the specificity mAbs in Fig 3C, except for mAb No. 9 (Lines 277 to 279). 

Answer 7.

We sincerely thank the reviewer for raising this concern. As commented in Line 355-358 in discussion session, the three mAbs obtained in this study look to be structure-recognition antibody. This type of antibodies can not recognize unfolded proteins (Matsuoka et al., 2010). So, we think only weak signals were detected by immunoblot analyses. We moved the immunoblot data (Fig 1E and 3C) to supporting information. In addition, we added following sentence in discussion section (Line 358-359) for explanation of structure-recognition antibody.

“This type of antibodies is reported to be hard to recognize unfolded proteins [22].”

22. Matsuoka K, Komori H, Nose M, Endo Y, Sawasaki T. Screening method for autoantigen proteins using the biotinylated protein library produced by wheat cell-free synthesis. J Proteome Res 2010; 9: 4264-4273.

Comment 8.

Provide reference for statements in the Introduction section (lines 67 to 70).

Answer 8.

We thank the reviewer for bringing up these constructive comments. In response to the reviewer’s comment, we added three references described below.

1. Iwanami T, Koizumi M, Ieki H. Diversity of properties among satsuma dwarf virus and related viruses. Ann Phytopathol Soc Jpn 1993;59: 642-650.

2. Shimizu S, Miyoshi T, Maruyama A, Ikegami M. Detection of satsuma dwarf virus (SDV) and its related viruses by ELISA using antiserum against SDV. Jpn J Plant Pathol 2002;68: 236-237.

3. Shimizu S, Miyoshi T, Tachibana Y. Susceptibility of citrus varieties to satsuma dwarf vires. Jpn J Plant Pathol 2007;73: 225.

Comment 9.

Perform statistical analysis for the ELISA data in Fig 4C, provide details on number of experiments including replicates involved. 

Answer 9.

We thank the reviewer for bringing up these constructive comments. In response to the reviewer’s comment, we performed statistical analysis in Fig 4C, and added sentences in legend of Fig 4C (Line 333-335) as described below.

“The data are shown as means ¬¬± standard deviations (indicated with error bars) from three independent experiments. Different letters indicate significant differences (P < 0.05, Tukey test).”

In addition, we replaced the sentences “The result showed that signals were 1.5 to 2.5 times higher in the CiMV and NIMV sections of mAbs No. 9 and 20 (Fig 4C) whereas equivalent or weaker signals were observed in all sections of the No. 4 mAb. In the SDV section of mAb No. 9 and 20, early equivalent signals were observed.” with “The result showed that signals were significantly higher in the CiMV and NIMV sections of mAbs No. 9 and 20 whereas equivalent or weaker signals were observed in the SDV section of mAb No. 9 and 20 (Fig 4C). In all sections of mAb No. 4, equivalent or weaker signals comparing with virus-free sample were observed.” in the results section (Line 306-310).

Comment 10.

How many experiments involved in Fig 4D? Provide quantitated dot-blot analysis based on staining density – this analysis help better understand sensitivity of mAb No. 20.

Answer 10.

We thank the reviewer for bringing up these constructive comments. In response to the reviewer’s comment, we performed multiple dot-blot analysis. The results were quantitated and statistical analysis were performed. These results were added to Fig 4D.

---

## [Decision Letter · Decision Letter 1]

3 Feb 2020

Production of a rabbit monoclonal antibody for highly sensitive detection of citrus mosaic virus and related viruses

PONE-D-19-23810R1

Dear Dr. Nozawa,

We are pleased to inform you that your manuscript has been judged scientifically suitable for publication and will be formally accepted for publication once it complies with all outstanding technical requirements.

With kind regards,

Joseph J Barchi

Academic Editor

PLOS ONE

Additional Editor Comments (optional):

Reviewers' comments:

Reviewer's Responses to Questions

**Comments to the Author**

1. If the authors have adequately addressed your comments raised in a previous round of review and you feel that this manuscript is now acceptable for publication, you may indicate that here to bypass the “Comments to the Author” section, enter your conflict of interest statement in the “Confidential to Editor” section, and submit your "Accept" recommendation.

Reviewer #1: All comments have been addressed

Reviewer #2: All comments have been addressed

2. Is the manuscript technically sound, and do the data support the conclusions?

Reviewer #1: (No Response)

Reviewer #2: Yes

3. Has the statistical analysis been performed appropriately and rigorously? 

Reviewer #1: (No Response)

Reviewer #2: Yes

4. Have the authors made all data underlying the findings in their manuscript fully available?

Reviewer #1: (No Response)

Reviewer #2: Yes

5. Is the manuscript presented in an intelligible fashion and written in standard English?

Reviewer #1: (No Response)

Reviewer #2: Yes

6. Review Comments to the Author

Reviewer #1: (No Response)

Reviewer #2: (No Response)

7. PLOS authors have the option to publish the peer review history of their article (what does this mean?). If published, this will include your full peer review and any attached files.

Reviewer #1: No

Reviewer #2: No

---

## [Editor Report · Acceptance letter]

26 Mar 2020

PONE-D-19-23810R1 

Production of a rabbit monoclonal antibody for highly sensitive detection of citrus mosaic virus and related viruses 

Dear Dr. Nozawa:

I am pleased to inform you that your manuscript has been deemed suitable for publication in PLOS ONE. Congratulations! Your manuscript is now with our production department. 

With kind regards,

on behalf of

Dr. Joseph J Barchi 

Academic Editor

PLOS ONE